# OpenReview forum: "Attention as a Hypernetwork"
_ICLR.cc/2025/Conference — ICLR 2025 Oral_

### Official Review · Reviewer_yrk8 · 2024-11-01

**Soundness:** 3
**Presentation:** 3
**Contribution:** 2
**Rating:** 8
**Confidence:** 4

**Summary:**

This paper presents an interpretation of multi-head attention as a "hypernetwork", in the sense of Ha et al. (2017)---a network that generates weights that parameterize another network. The main insight comes from writing MHA as: $\sum_{h=1}^{H} W_h^{out}\sum_{k=1}^{K} a_{h,q,k} W_h^{val} x_k = \sum_{k=1}^{T} \left(\sum_{h=1}^{H} a_{h,q,k} W_h^{out} W_h^{val}\right) x_k$, and interpreting the inner summation in the RHS as the parameterization of the "value network". This comes from swapping the summation over heads $h$ with the summation over keys in the typical formulation of attention. The authors argue that this can give rise to some notion of "compositionality" and OOD generalization. While the value network here is parameterized as a linear combination of linear maps, the authors use this interpretation to propose a variant of linear attention where the value network is instead a two-layer MLP, with each linear map in the MLP being parameterized by the hypernetwork. The paper then explores two experimental settings: the first is based on ICL on "fuzzy logic functions" and the second is based on a symbolic version of Raven's progressive matrices. The authors show that networks with linear attention and softmax attention achieve some level of OOD generalization on these tasks, and that their proposed HYLA model achieves further improved performance.

**Strengths:**

- The question of how compositional generalization arises in neural networks, and Transformers specifically, is of broad interest to the ML research community. I believe that the interpretation proposed in this paper is a meaningful contribution to this question.
- The paper is well-written, well-structured, and clear.
- To my knowledge, the interpretation of combinations of heads as a hypernetwork is novel.
- The proposal of the HYLA attention mechanism is well-motivated and makes sense given the presented hypernetwork interpretation of MHA.
- The experimental results provide some empirical support for the general claims made in the paper, with HYLA performing favorably in terms of compositional generalization on the tasks considered.
- I found the figures showing the correlation between the latent codes and the target label (i.e., 2B and 5A) compelling, making the argument that some notion of compositionality is made possible by having multiple heads and

**Weaknesses:**

- The interpretation of MHA as a hypernetwork through re-ordering the summation over heads and keys makes sense in the case of linear attention, but seems less so in the case of softmax attention. In the case of softmax attention, the value network is a linear combination over heads, but with the normalization being over keys rather than heads. This doesn't fit into the claim that the value network is merely query-key dependent; rather it is a function of the entire sequence. Can the authors comment on this? You focus more on linear attention in the paper, and perhaps this is the reason. But, can you share more of your thinking about this?
- The idea of a value network being input-dependent seems closely related to Mixtures-of-Experts. However this is not directly addressed in the paper. Can you comment on this? In particular, is there a type of function that HYLA can represent that standard MHA followed by a MoE-MLP could not? Experimentally, how would the latter perform compared to HYLA? Would it be better or worse at compositional generalization?
- Although the experiments are clear, the scenarios where compositionality and the advantage of HYLA are demonstrated are somewhat restricted in scope. On complex real-world tasks such as language modeling, HYLA performs worse compared to Softmax attention (this is again related to HYLA being a variant of linear attention).

See also the questions below.

Overall, I think this is a nice research paper, with meaningful contributions, and I look forward to discussion with the authors.

**Questions:**

- How does this interpretation of MHA as a hypernetwork relate to the "residual stream" view, and related concepts on attentional circuits in previous mechanistic interpretability work? Does it have any implications on simple circuits like, say, induction heads?
- How do MoE models compare in these compositional generalization experiments?
- Typically, the value network in MHA is thought of as merely projecting out a certain component of the context token to be retrieved and incorporated into the query token. The value network is not intended to perform any complex processing, that is the job of the MLP that immediately follows. However, the interpretation of MHA as a hypernetwork seems to hinge on the hypothesis that the value network ought to be doing some more complex processing. Can you comment on this distinction? Why would we want the value network to perform complex processing rather than having that be performed by the MLP?
- Under the hypernetwork view, the "value network" has a pretty specific parameterization as a query/key-dependent linear combination of linear maps. Can you comment on why this type of parameterization would make sense? For example, is there some simple construction of a network where there is an interpretable view of the class of functions this could capture?
- In lines 370-374, the authors discuss the difference between standard RPM problems with visual inputs and their simplified version with symbolic input, arguing that the main source of difficulty in RPM tasks is the problem of finding correspondences of rules to feature dimensions. I found this unconvincing, however, and I hope the authors can expand. In particular, it seems to me that this argument relies on the assumption that the visual representation learning step of discovering the right symbolic abstractions is easy, citing work in cognitive science. But what might be easy for humans (who have in-built abilities through millions of years of evolution) may not be easy for machines. Learning the full visual RPM problem end-to-end is surely significantly more difficult for a machine learning system, no? Overall, I think it's unnecessary to try to make this argument. Yes, the symbolic RPM problem is easier, but it is still interesting; we don't need to argue that it captures all the interesting aspects of the full visual RPM problem.

---

> ### Author Response · Authors · 2024-11-21
>
> We are grateful for your positive feedback on the quality of our paper's writing and its contribution towards understanding compositional generalization in neural networks. Based on your comments and suggestions, we have conducted additional experiments on MoEs and revised parts of the manuscript. In the following we answer your points one-by-one hoping to clarify any remaining concerns.
>
> > The interpretation of MHA as a hypernetwork through re-ordering the summation over heads and keys makes sense in the case of linear attention, but seems less so in the case of softmax attention. [...]
>
> The interaction with the softmax is indeed an interesting point to consider. First of all we would like to clarify that we are not claiming that the value network is merely query-key dependent but rather query-key specific in the sense that from the hypernetwork perspective each query-key pair has a different latent code. We apologize if this was not clear enough in the manuscript - we have tried to improve the clarity of this point but please let us know if there are particular passages in the paper that need further improvement. As you correctly point out, the softmax introduces an interaction across keys that makes the key-query specific latents dependent on the whole sequence. Specifically it creates additional competition across individual entries of the latent code before this code is used to parameterize the value network. Our latent code analysis reveals that even with this competition among components a structured latent code can emerge that allows decoding operations performed by the network (see Figure 5B+F specifically).
>
> > The idea of a value network being input-dependent seems closely related to Mixtures-of-Experts. [...]
>
> This is a very good question. One broader point to clarify in this context is that in principle multiple transformer blocks with standard MHA and standard MLPs can already emulate a single transformer block with HYLA. Intuitively, such a construction can be obtained by leveraging the nonlinearity of the MLP layer of the first block to function as the additional nonlinearity of HYLA. We primarily introduce HYLA as an experimental device that encourages the use of the hypernetwork circuit motif in order to evaluate whether it has the hypothesized effect on compositional generalization. Experimentally we observe in Figure 4 that at sufficient scale standard attention mechanisms can achieve comparable compositional generalization performance to HYLA. That being said, the idea that MoEs might similarly allow to compartmentalize and compose knowledge is interesting and we have added this analysis (see next response).
>
> > How do MoE models compare in these compositional generalization experiments?
>
> We have conducted these additional experiments based on your suggestions and added them to Appendix A1. We replaced the MLP layers with MoEs with E=5 experts of the size of the original MLP layer and a router that sparsely activates the top k=2 experts, i.e. each MoE has ~5x more parameters than the original MLP block and ~2x the number of active parameters (discounting the parameters added by the router),
>
> $$f_{\mathrm{MoE}}({x})= \sum_{i=1}^{E}\mathrm{Softmax}(\mathrm{Top\_k}({W}^{\text{gate}}{x}))\cdot{W}^{\mathrm{up}}_i\phi({W}^{\mathrm{down}}_i{x}).$$
>
> This generally leads to a slight improvement in terms of OOD performance across attention variants which is in line with our observations increasing the capacity of the models by introducing more depth/width shown in Figure 4 but does not point to these types of MoEs specifically improving compositional generalization.

---

> > ### Author Response · Authors · 2024-11-21
> >
> > > Although the experiments are clear, the scenarios where compositionality and the advantage of HYLA are demonstrated are somewhat restricted in scope. On complex real-world tasks such as language modeling, HYLA performs worse compared to Softmax attention (this is again related to HYLA being a variant of linear attention).
> >
> > We would like to clarify that we are not positioning HYLA as a replacement for softmax attention but rather as an experimental device that specifically encourages the use of the hypernetwork mechanism of multi-head attention. It is known that the softmax allows to implement circuit motifs that are very powerful on language modeling such as, for instance, the induction heads you mention [1] which show that the softmax can copy information and solve binding/associative recall problems on which linear attention struggles [2,3]. However, we believe the fact that modifying linear attention in a way that reinforces the hypernetwork mechanism reduces the gap to softmax attention provides evidence that the hypernetwork mechanism can play a practical role for understanding multi-head attention in language modeling.
> >
> > [1] Olsson et al. "In-context learning and induction heads." Transformer Circuits (2022).
> >
> > [2] Arora et al., Zoology: Measuring and Improving Recall in Efficient Language Models (2023)
> >
> > [3] Schlag et al., Linear Transformers Are Secretly Fast Weight Programmers (2021)
> >
> >
> > > How does this interpretation of MHA as a hypernetwork relate to the "residual stream" view, and related concepts on attentional circuits in previous mechanistic interpretability work? Does it have any implications on simple circuits like, say, induction heads?
> >
> > We think of the hypernetwork interpretation as one possible class of attentional circuits MHA layers can implement. The flexibility of MHA allows for other classes of attentional circuits such as the induction heads you mention. The main goal of our experiments is therefore to provide evidence that the hypernetwork circuit motif can arise in practice. In those cases, it provides a new angle to mechanistically interpret functions performed by a MHA layer through the latent code of the hypernetwork and the associated functions implemented by the corresponding value networks.
> >
> > > Typically, the value network in MHA is thought of as merely projecting out a certain component of the context token to be retrieved and incorporated into the query token. The value network is not intended to perform any complex processing, that is the job of the MLP that immediately follows. However, the interpretation of MHA as a hypernetwork seems to hinge on the hypothesis that the value network ought to be doing some more complex processing. Can you comment on this distinction? Why would we want the value network to perform complex processing rather than having that be performed by the MLP?
> >
> > We are not claiming that the architecture of the transformer should be changed such that the value network performs more complex processing. The point we are trying to make is that the flexibility of MHA allows it to implement the hypernetwork circuit motif. By making the value network nonlinear, we strengthen the hypernetwork mechanism in order to show its influence on compositional generalization where it can indeed be beneficial since it allows specific, reusable operations to perform more complex processing.
> >
> > > Under the hypernetwork view, the "value network" has a pretty specific parameterization as a query/key-dependent linear combination of linear maps. Can you comment on why this type of parameterization would make sense? For example, is there some simple construction of a network where there is an interpretable view of the class of functions this could capture?
> >
> > From the hypernetwork perspective there are two neural network components to interpret: the hypernetwork generating the weights and the value network that is parameterized by these weights. Another way of saying “linear combination of linear maps” in these terms is that standard MHA uses a linear hypernetwork to generate a linear value network. While in principle nonlinear hypernetworks are more expressive, prior work has found that linear hypernetworks can achieve similar performance [4].
> >
> > [4] Schug et al. “Discovering modular solutions that generalize compositionally.” (2024)
> >
> > > [...] Yes, the symbolic RPM problem is easier, but it is still interesting; we don't need to argue that it captures all the interesting aspects of the full visual RPM problem.
> >
> > Thank you for raising this point, we agree with your assessment and have clarified this passage in the revised manuscript accordingly.

---

> > ### Comment · Reviewer_yrk8 · 2024-11-22
> >
> > Dear authors,
> >
> > Thank you very much for your detailed responses to all reviewers. I tried to take a look at your responses to each review and the additions made to the paper.
> >
> > **On hypernetworks in linear vs softmax attention.**
> >
> > > First of all we would like to clarify that we are not claiming that the value network is merely query-key dependent but rather query-key specific in the sense that from the hypernetwork perspective each query-key pair has a different latent code
> >
> > I'm not sure what you mean by this. Please let me know if you believe I've misunderstood something. What's the difference between "query-key dependent" and "query-key specific"? The value network applied to each element in the context is a linear combination of multiple value networks, and the weights of this linear combination are a function of the query and the key. (I don't see how a value network could be "query-key dependent" but not "query-key specific".)
> >
> > More generally, I take your point that Figures 5B+F show some structure in the "latent codes" (i.e., the vectors $(a_{q,k,h})_h$, but I'm not sure I see that this hypernetwork perspective is the best explanation of this in the case of softmax attention. It seems to me that this can be explained by the "classical" interpretation of different heads in multi-head attention corresponding to different "selection criteria" (perhaps with different heads involved in different circuits).
> >
> > I do think that the connection between linear attention and hypernetworks is neat. But I view it as another possible lens through which to view the same mechanism (with other interpretations also being valid and useful). In the case of softmax attention, the hypernetwork argument does not seem to apply (at least nearly as cleanly), and other interpretations seem more accurate/conceptually useful. Please let me know if you believe I've missed something.
> >
> > As I said, although this is a limitation, I do think the linear attention case is interesting in itself.
> >
> > **Relation to MoE**
> > > We have conducted these additional experiments based on your suggestions and added them to Appendix A1.
> >
> > Thank you for this, this is helpful!
> >
> > **HYLA as an experimental device**
> >
> > > We would like to clarify that we are not positioning HYLA as a replacement for softmax attention but rather as an experimental device that specifically encourages the use of the hypernetwork mechanism of multi-head attention.
> >
> > If I understand your argument correctly, you're framing HYLA as an "experimental device", intended mainly to study hypernetwork effects in standard multi-head attention, and how that may give rise to compositional generalization. But HYLA deviates significantly from standard linear MHA, and your experimental results show that the "hypernetwork effects" in standard MHA are relatively weak (i.e., the generalization performance is significantly worse). The fact that HYLA operates as a hypernetwork does not imply that standard MHA operates as a hypernetwork.
> >
> > > We are not claiming that the architecture of the transformer should be changed such that the value network performs more complex processing. The point we are trying to make is that the flexibility of MHA allows it to implement the hypernetwork circuit motif.
> >
> > I'm having trouble understanding how this argument relates to my question. HYLA does have more complex processing in the value hypernetwork, which is significantly different from the weak type of "hypernetwork" in linear MHA.

---

> > > ### Author Response · Authors · 2024-11-22
> > >
> > > Thank you for engaging in an important discussion with us and giving as an opportunity to further clarify how we believe the hypernetwork perspective allows us to interpret the collection of multiple heads to form a circuit together not only for linear attention but also for softmax attention.
> > >
> > > To make the argument easier to follow, we briefly restate the mathematical formulation of multi-head attention as a hypernetwork. For a given a query $q$, we have
> > >
> > > $$
> > > \mathrm{MH}A_q(\mathrm{X}) = \sum_{k=1}^T \left(\sum_{h=1}^H a_{h,q,k} W^{\mathrm{out}}_h W^{\mathrm{value}}_h \right) {x}_k
> > > $$
> > >
> > > > I'm not sure what you mean by this. Please let me know if you believe I've misunderstood something. What's the difference between "query-key dependent" and "query-key specific"? The value network applied to each element in the context is a linear combination of multiple value networks, and the weights of this linear combination are a function of the query and the key. (I don't see how a value network could be "query-key dependent" but not "query-key specific".)
> > >
> > > We apologize for not being clear on this but all we mean to say with “the latent code is query-key specific” is that in the equation above for each query index and key index, there is one vector $[a_{1,q,k}, a_{2,q,k}, \dots, a_{H,q,k}]$ that we interpret as the latent code of a hypernetwork. As you correctly pointed out, the fact that there is one latent code per query-key pair does not mean that the latent code only functionally depends on this query-key pair. The softmax normalization across keys for example introduces a functional dependence on the whole sequence.
> > >
> > > > More generally, I take your point that Figures 5B+F show some structure in the "latent codes" (i.e., the vectors $a_{h,q,k}$, but I'm not sure I see that this hypernetwork perspective is the best explanation of this in the case of softmax attention. It seems to me that this can be explained by the "classical" interpretation of different heads in multi-head attention corresponding to different "selection criteria" (perhaps with different heads involved in different circuits).
> > > > I do think that the connection between linear attention and hypernetworks is neat. But I view it as another possible lens through which to view the same mechanism (with other interpretations also being valid and useful). In the case of softmax attention, the hypernetwork argument does not seem to apply (at least nearly as cleanly), and other interpretations seem more accurate/conceptually useful. Please let me know if you believe I've missed something.
> > >
> > > This is a really interesting and important point but we would like to emphasize that we do believe there is good evidence that also in the case of softmax attention the hypernetwork mechanism is a useful explanatory lens:
> > >
> > > Mathematically, the equivalence between the hypernetwork mechanism and multi-head attention does equally apply to softmax attention - it makes no assumption about attention being linear. That being said, we agree that a priori it is unclear if there is additional explanatory power in thinking of multiple heads as implementing a circuit together vs. each head simply forming their own circuit, that can be understood independently of other heads (but potentially still combined). It could be that the networks learned in practice use this “independent heads” mode, for example because the softmax competition across keys (and therefore across latent codes) creates sparse latent codes where only one head is active per query-key pair. However, in practice we find evidence that heads do “collaborate” and are better understood in conjunction. Figure 5B (also see Figure A7 for all layers of a deeper softmax transformer) shows that the latent codes form a highly structured code and figure 5F shows that this latent code not only visually matches the task operations but also allows to decode these operations. One additional piece of evidence to make this point is to make sure that each of these latent codes is not simply a one-hot vector, i.e. a single head implementing a circuit but multiple heads that consistently work together to implement a circuit. We have added Figure A8B to the appendix that shows that the average latent code per rule typically includes multiple heads being active for a given rule. Figure A8A in addition shows a random batch of individual, non-averaged latent codes to make sure the co-occurence of heads happens on a per example basis and is not the result of averaging.
> > >
> > > Taken together we believe this provides evidence for the hypothesis that heads “collaborate” to implement circuits that can be compactly identified through the latent code.

---

> > > > ### Author Response · Authors · 2024-11-22
> > > >
> > > > > If I understand your argument correctly, you're framing HYLA as an "experimental device", intended mainly to study hypernetwork effects in standard multi-head attention, and how that may give rise to compositional generalization. But HYLA deviates significantly from standard linear MHA, [...]
> > > >
> > > > In addition to the latent code analysis, we use HYLA as a way of studying what happens when the hypernetwork mechanism is reinforced in an attempt to understand how this mechanism in particular affects compositional generalization. It combines three modifications: (1) normalization of the attention scores across the head indices using RMSHead, (2) applying the attention-weighted sum operation also to the output projection and (3) inserting a nonlinearity in the value network. Each of these modifications can be seen as a separate intervention towards reinforcing the hypernetwork mechanism. Table A1 shows that each of these modifications by themselves have a positive effect on compositional generalization (of course to varying degrees), so the effects we are seeing also appear to be present for less deviating variants.
> > > >
> > > > > [...] and your experimental results show that the "hypernetwork effects" in standard MHA are relatively weak (i.e., the generalization performance is significantly worse). The fact that HYLA operates as a hypernetwork does not imply that standard MHA operates as a hypernetwork.
> > > >
> > > > We would like to also emphasize that as standard MHA is scaled up in terms of width and depth the generalization performance is on par with reinforcing the hypernetwork mechanism through HYLA as shown in Figure 4. In addition, as outlined before we find a highly structured latent code in this case as well. So we do think the “hypernetwork effects” are also observed in softmax MHA.
> > > >
> > > > > I'm having trouble understanding how this argument relates to my question. HYLA does have more complex processing in the value hypernetwork, which is significantly different from the weak type of "hypernetwork" in linear MHA.
> > > >
> > > > Apologies, some formatting got lost when we entered our response. This was meant to specifically answer the question “Why would we _want_ the value network to perform complex processing rather than having that be performed by the MLP?”. We were trying to clarify that we are not saying it _should_ replace MHA in general, but as outlined in the response to one of the previous points is one of the ways of trying to experimentally separate the effects of the hypernetwork mechanism on compositional generalization.
> > > >
> > > > We hope that our response helped convince you that there is evidence to believe the hypernetwork mechanism is a useful explanatory lens to understand how multiple heads form circuits together that can also be observed in standard transformers based on softmax MHA.

---

> ### Comment · Reviewer_yrk8 · 2024-11-24
>
> Thank you. I appreciate the continued discussion and your detailed responses to my questions and concerns.
>
> I have raised my score in appreciation of the authors' efforts in improving the clarity of the presentation and the additional experimental results, which have addressed some of my concerns.
>
> However, on a final note, I'm not entirely convinced of the argument that it is useful to view softmax attention as a hypernetwork.  Ultimately, all this seems to build down to is a simple question---Is it more useful to understand MHA as
> $$\sum_{h} \sum_{k} \alpha_{qk}^h \\, W_{ov}^h \\, x_k \quad \text{(Standard Interpretation)}$$
> or as
> $$\sum_{k} \sum_{h} \alpha_{qk}^h \\, W_{ov}^h \\, x_k \quad \text{(Hypernetwork Interpretation)}$$
>
> This seems to be what you refer to as the "mathematical equivalence". But this isn't a deep mathematical derivation, it's just swapping the order of the summation. This might give rise to a useful interpretation (which I believe it does in the case of linear attention, as you demonstrate in the paper), but it is just an interpretation of the same underlying operation.
>
> In the hypernetwork interpretation, you interpret $ \sum_{h} \alpha_{qk}^h W_{ov}^h$ as a hypernetwork, because it is a linear combination of linear maps, with the linear weights determined by the input. This is a neat observation in the case of linear attention, where the two summations are intuitively exchangeable. But when you have (softmax) normalization over $k$, the hypernetwork interpretation obfuscates this crucial aspect and gives the wrong idea. You can stretch the definition of a "hypernetwork" to allow for this kind of scenario, but it is not clear that this continues to be a useful conceptual framework in that case.
>
> > we agree that a priori it is unclear if there is additional explanatory power in thinking of multiple heads as implementing a circuit together vs. each head simply forming their own circuit, that can be understood independently of other heads (but potentially still combined).
>
> I don't think that the hypernetwork interpretation is necessary to understand multiple heads as implementing a circuit together. There is clearly an interaction between heads since they add up in their contribution to the residual stream, and the attention patterns of the different heads are a function of the same underlying input. I don't think that one needs to adopt the hypernetwork interpretation to understand how heads "collaborate". If you'd like to argue for the hypernetwork interpretation, I think you would need to demonstrate that this very particular way of parameterizing the linear weights that define the hypernetwork (including the softmax normalization over keys) gives rise to some interpretable operation. In the paper, you show that the latent codes have structure, but you don't interpret the exact "hypernetwork" that was learned.
>
> I do believe that the exploration of the "latent code" $[\alpha_{qk}^1, ..., \alpha_{qk}^H]$, and interpreting its correlation to the target is interesting. I had not seen it before. But again, even under the standard "$\sum_{h} \sum_{k} \alpha_{qk}^h \\, W_{ov}^h \\, x_k $" interpretation, you would still expect some structure in the latent code (since different heads will activate in different places in an input-dependent way). You don't necessarily need the hypernetwork interpretation (which obfuscates the normalization over keys, which is often crucial).
>
> ---
>
> My final suggestion to the authors, in case it's useful, is to consider revising the framing of the paper to more explicitly explain the purpose of proposing HYLA (i.e., as an "experimental device", and their argument about where the hypernetwork interpretation is or is not useful. I hope some of my feedback was helpful.

---

> > ### Author Response · Authors · 2024-11-25
> >
> > Thank you for your constructive feedback throughout the discussion period, we found it very useful. We appreciate that you acknowledge the improvements made and that you have hence updated your score recommending our paper for clear acceptance.
> >
> > Please allow us to briefly clarify one important point: The reformulation of MHA as a hypernetwork consists of two parts, equation (4) which you state above but importantly also equation (5), namely:
> > $$
> > \sum_k \sum_h a_{h,q,k} W_h^{out} W_h^{value} x_k = \sum_k W_{q,k} x_k \qquad \text{(Hypernetwork interpretation)}
> > $$
> > This part is important since it suggests that we can think of a single operation per key-query pair instead of reasoning about $H$  independent circuits. These operations are reusable/shared through one common linear hypernetwork that takes as input the latent code.
> >
> > > This is a neat observation in the case of linear attention, where the two summations are intuitively exchangeable. But when you have (softmax) normalization over , the hypernetwork interpretation obfuscates this crucial aspect and gives the wrong idea.  You can stretch the definition of a "hypernetwork" to allow for this kind of scenario, but it is not clear that this continues to be a useful conceptual framework in that case.
> >
> > Mathematically this is how a hypernetwork is defined, it is a neural network that generates the weights for another neural network, so we do not believe there is any ambiguity in referring to the mechanism in this formula as a hypernetwork. The two summations are exchangeable both in the case of linear attention  and softmax attention. But of course we agree that this does not imply that this interpretation is always the best to understand the circuits of a trained transformer - it is one possible “mode of operation” and we provide evidence that this mode occurs in practice (as reiterated in our previous response). But we agree with you that other “modes” exist where the softmax plays a key role such as in binding and associative recall problems.
> >
> > > My final suggestion to the authors, in case it's useful, is to consider revising the framing of the paper to more explicitly explain the purpose of proposing HYLA (i.e., as an "experimental device", and their argument about where the hypernetwork interpretation is or is not useful. I hope some of my feedback was helpful.
> >
> > We will try to improve this important point further, thank you for raising it.

---

### Official Review · Reviewer_Rtsz · 2024-11-03

**Soundness:** 3
**Presentation:** 2
**Contribution:** 2
**Rating:** 8
**Confidence:** 2

**Summary:**

The paper investigates how Transformers generalize to new, unseen task compositions by reformulating multi-head attention as a hypernetwork—a model that reconfigures another model's computation based on data. This study reveals that a compact, low-dimensional latent code within the attention mechanism specifies key-query operations, which can generalize compositional tasks even if those specific combinations weren’t encountered during training.

**Strengths:**

1. The paper provides an innovative perspective by reformulating multi-head attention as a hypernetwork, a novel conceptual framework that opens up new avenues for understanding attention mechanisms in neural networks.

2. The paper is well-structured, with clear explanations of its objectives, methods, and findings.

**Weaknesses:**

1. The paper makes a strong case for viewing multi-head attention as a hypernetwork, but it lacks a formal theoretical analysis showing why this approach specifically enhances compositional generalization over traditional attention mechanisms. Although the empirical results support the hypernetwork framing, a deeper theoretical exploration would add rigor and clarity.

2. The paper primarily contrasts the proposed approach with standard softmax and linear attention but lacks comparisons with other recent methods that enhance compositional generalization in Transformers.

3. While SRAVEN offers a modular benchmark for evaluating compositional generalization, it may not fully reflect the demands of compositional reasoning in language or perception tasks, which involve a broader range of dependencies and symbolic variations.

4. Although the proposed Hypernetwork Linear Attention (HYLA) and hypernetwork framing show promising results, there is limited discussion on scalability and practical implementation challenges, especially as the model size increases. For instance, it remains unclear whether the approach would scale effectively in large language models (LLMs) or in high-dimensional data contexts.

5. The authors posit that the structured latent code in multi-head attention serves as a reusable resource for different compositional tasks. While the paper provides clustering analyses, there is limited interpretability regarding how the latent codes correlate with specific task elements or subtasks.

**Questions:**

1. Could you provide a more formal explanation or theoretical basis for why framing multi-head attention as a hypernetwork enhances compositional generalization? Specifically, what is the role of the latent code in enabling the network to handle unseen task compositions?

2. Given the computational demands of Transformers in large-scale applications, would HYLA introduce significant memory or efficiency constraints, and if so, how might these be addressed?

3. Have you considered conducting deeper analyses, such as probing or interpretability methods, better to understand the latent codes’ alignment with specific subtasks?

---

> ### Author Response · Authors · 2024-11-21
>
> We are happy to hear that you believe our paper offers an innovative perspective that opens up new avenues for understanding attention mechanisms in neural networks. It is in this light that we introduce HYLA, namely as an experimental device to improve our understanding of attention. We believe that some of your concerns might be a result of considering HYLA as a suggested replacement for the attention mechanism in transformers to improve their performance but this is not what we propose. We apologize for this potential misunderstanding. In the following we answer your points more specifically, hoping that you reconsider your overall rating of our submission in light of this clarification.
>
> > The paper makes a strong case for viewing multi-head attention as a hypernetwork, but it lacks a formal theoretical analysis showing why this approach specifically enhances compositional generalization over traditional attention mechanisms.
>
> In section 2.1 we provide a mathematical derivation that links multi-head attention to hypernetworks. We would like to clarify that this derivation applies to traditional attention mechanisms (e.g. multi-head softmax and linear attention) and is not by itself a separate approach to improve compositional generalization.
> Prior work has theoretically and empirically shown that hypernetworks support compositional generalization [1]. We apologize that this point has not been made more explicit in the original submission and have revised section 2.1 to clarify this. With the theoretical link between multi-head attention, hypernetworks and compositional generalization in place, we design our experiments to verify the predictions made by this theoretical framework, for instance showing that the latent code of the hypernetwork is predictive of the operations implemented by multi-head attention.
>
> [1] Schug et al. “Discovering modular solutions that generalize compositionally.” (2024)
>
> > The paper primarily contrasts the proposed approach with standard softmax and linear attention but lacks comparisons with other recent methods that enhance compositional generalization in Transformers.
>
> Given that our goal is to understand to what extent the hypernetwork mechanism explains network computations of learned transformers, the introduction of HYLA serves primarily as an experimental device to examine the hypothesis that the intrinsic hypernetwork of multi-head attention is what supports compositional generalization on abstract reasoning tasks that require recombining previously acquired operations. We attempt to clarify this part in the abstract (line 16-19). Furthermore as we show in Figure 4, at sufficient scale, linear and softmax attention can similarly compositionally generalize. Importantly, their internal workings can be understood through the hypernetwork mechanism as for instance the analyses in Figure 5 aim to show.
>
> > While SRAVEN offers a modular benchmark for evaluating compositional generalization, it may not fully reflect the demands of compositional reasoning in language or perception tasks, which involve a broader range of dependencies and symbolic variations.
>
> We consider abstract reasoning problems such as SRAVEN and the fuzzy logic task since this gives us precise control over the problem compositions encountered during training and evaluation and allows us to verify specific hypotheses pertaining to the mathematical equivalence of multi-head attention to a hypernetwork - for example the ability to decode the operations performed by the network based on the latent code in Figure 5F. We furthermore conduct language modeling experiments with 50M parameter models trained on 150B tokens. We find that also in this setting the experimental intervention of reinforcing the hypernetwork mechanism of standard multi-head linear attention through HYLA leads to a noticeable improvement in performance performing closely to softmax attention. This is noteworthy since it is in a setting where the softmax is known to be important due to its role in binding and associative recall problems on which linear attention methods typically struggle [2,3,4].
>
> [2] Arora et al., Zoology: Measuring and Improving Recall in Efficient Language Models (2023)
>
> [3] Schlag et al., Linear Transformers Are Secretly Fast Weight Programmers (2021)
>
> [4] Olsson et al., In-Context Learning and Induction Heads (2022)

---

> > ### Author Response · Authors · 2024-11-21
> >
> > > Although the proposed Hypernetwork Linear Attention (HYLA) and hypernetwork framing show promising results, there is limited discussion on scalability and practical implementation challenges, especially as the model size increases. For instance, it remains unclear whether the approach would scale effectively in large language models (LLMs) or in high-dimensional data contexts.
> >
> > Please see our response above that we do not claim that HYLA should be used to replace attention mechanisms but is rather an experimental device for understanding such mechanisms that are already successfully scaled up. Our general derivation holds for standard multi-head attention.
> >
> > > The authors posit that the structured latent code in multi-head attention serves as a reusable resource for different compositional tasks. While the paper provides clustering analyses, there is limited interpretability regarding how the latent codes correlate with specific task elements or subtasks.
> >
> > We address this point specifically in the last paragraph of section 4.4 and the corresponding Figure 5F where we train a linear decoder to predict the ground-truth SRAVEN rule based on the latent code at the final query token of training tasks and evaluate it on unseen OOD tasks revealing that the latent code is predictive of the implemented rule. We perform a similar decoding analysis for the fuzzy logic task in section 3.2 and the corresponding Figure 2B.

---

### Official Review · Reviewer_47bR · 2024-11-03

**Soundness:** 3
**Presentation:** 3
**Contribution:** 3
**Rating:** 6
**Confidence:** 2

**Summary:**

The paper aims at understanding the ability for compositional generalization for the transformer networks. The idea is to view the multihead attention module as a hypernetwork, where a composable, low-dimensional latent code specifies key-query specific operations. Empirically, the paper finds that using non-linear value networks for the hypernetwork (i.e., the multihead attention module) leads to improved generalization ability. The paper further introduces a symbolic RAVEN test for fine-grained evaluations of compositional generalization ability.

**Strengths:**

- The idea of viewing multi-head attention as a hypernetwork is interesting.
- The experiments on a controlled domain such as symbolic RAVEN test are clear and thorough.
- The research problem is of great interest to the ML community.
- The paper is well-written and well-organized.

**Weaknesses:**

- I may be missing something. But the modification of the linear attention layer seems a little bit restricted. Will other non-linearity functions or $\sigma(\cdot)$ functions other than `RMSHead` also lead to improved compositional generalization ability? It would be great if the authors can comment on this and explain why this modification would work.
- I am admittedly not an expert of this domain, and would like to hear my colleague reviewers' thoughts on this.

**Questions:**

Please see the weaknesses section.

---

> ### Author Response · Authors · 2024-11-21
>
> Thank you for your encouraging review - we are glad that you found our experiments to be clear and the paper well written.
>
> > I may be missing something. But the modification of the linear attention layer seems a little bit restricted. Will other non-linearity functions or functions other than RMSHead also lead to improved compositional generalization ability? It would be great if the authors can comment on this and explain why this modification would work.
>
> Thank you for this suggestion, we have run additional experiments to test the influence of different nonlinearities on the performance of HYLA on SRAVEN (same task and model setup as in Table A1).
>
> | Nonlinearity | OOD accuracy     |
> |--------------|------------------|
> | gelu         | $58.92 \pm 1.50$ |
> | relu         | $69.13 \pm 1.90$ |
> | sigmoid      | $56.26 \pm 1.83$ |
> | silu         | $71.04 \pm 1.05$ |
> | tanh         | $54.87 \pm 0.26$ |
>
> These results show that the ReLU nonlinearity is generally a well performing choice but not the only one that can lead to improvements in compositional generalization. In addition to varying the nonlinearity of HYLA, we perform many model ablations in Appendix A1 to verify how the different modifications of HYLA affect compositional generalization. This includes testing a model that applies the RMSHead to linear attention and a variant that removes RMSHead from HYLA. Overall we find that all the modifications we introduce in HYLA over linear attention together benefit OOD performance on both SRAVEN and the fuzzy logic task.

---

> > ### Comment · Reviewer_47bR · 2024-11-25
> >
> > I would like to thank the authors for their detailed responses. My concerns have been addressed. Therefore I decide to keep my rating.

---

### Official Review · Reviewer_59h3 · 2024-11-03

**Soundness:** 4
**Presentation:** 3
**Contribution:** 3
**Rating:** 8
**Confidence:** 4

**Summary:**

Transforms have been shown to compositionally generalize in some situations and not in others. This paper presents a novel perspective of multi-head self-attention, a core component of transformers, which better exposes the module's capacity for representing and reusing compositional structure.

Under this perspective:
- The multi-head self-attention mechanism comprises of $\\text{num\\_queries}\\times\\text{num\\_keys}$ value networks that are generated by a hypernetwork (hypernet) parameterized by the value and output projection weights
- The inputs of the hypernet are $\\text{num\\_heads}$-dimensional "latent codes," each corresponding to a query-key pair. Relating to standard multi-head self-attention, these latent codes are obtained by indexing the attention matrix stacked for all heads at a specific key-value index.

With this perspective:
- The authors show that the latent codes exhibit structure that reflects the learned compositionality for the task, which can explain why transformers compositionally generalize when they do. For instance, the latent codes are empirically predictive of the subtasks the network performs on unseen compositions.
- The paper introduces the hypernetwork linear attention (HYLA), a drop-in replacement of standard multi-head self-attention, that modifies the hypernet-generated value network from linear to non-linear to make it more expressive. This modification also replaces the expensive global softmax normalizations per head with local key-query-specific normalizations across heads.

The authors introduce SRAVEN, a symbolic version of Raven Progressive Matrices human intelligence test, which gives them more precise control over the compositions during training and evaluation. Evaluation on SRAVEN shows clearer latent code structure and improved OOD accuracy of HYLA over softmax and linear attention.

Finally, the authors evaluate HYLA on a language modeling task, but it fails to outperform softmax attention here.

**Strengths:**

- The paper presents a novel perspective of multi-head attention as a hypernet. More specifically, the perspective that queries and keys dynamically create a function to apply to the values is a straightforward interpretation of the attention mechanism, but the proposed perspective further reveals where the learned compositional structure is potentially exhibited.
- The core technical contributions of the "multi-head self-attention as a hypernetwork" perspective and HYLA are presented very clearly.
- The proposed method HYLA is a novel drop-in replacement for standard multi-head self-attention.
- The authors clearly show emergent compositionality in the "latent codes" for the intrinsic hypernets of multi-head self-attention and HYLA.

**Weaknesses:**

The paper seems to consider the special case of self-attention:
- The proposed perspective of attention as hypernet is presented with the assumption of self-attention, which is a special case of attention (e.g., cross-attention is more general). While the assumption of self-attention can be seen in figures and equations, it is only explicitly mentioned in the text at line 114. Without clarifying that this perspective and the proposed modification (HYLA) consider the special case of self-attention, the paper can be seen as over-claiming its contribution. The paper's title, abstract, and introduction should clarify this constraint. Alternatively, the paper should clarify how its formulations work in more general cases of attention and if their experiments are limited to the particular case of self-attention.

Clarity and connection of multi-head to compositionality:
- The necessity and significance of the "multi-head" aspect for compositional generalization is underplayed. A relevant discussion is found in the first paragraph of Section 6, which could be discussed earlier in the paper to set up a clear connection to compositionality.
- The introduction of the paper is quite dense. After a clear flow of the first two paragraphs, it abruptly gets into too many technical specifics and jargon in the third paragraph that is difficult to parse without the context of the rest of the paper. Perhaps a less technical introduction of the paper's perspective and modification, with a motivation of how these relate to compositional generalization, can set up clearer expectations for what is to come and why it matters (maybe the Summary of this review can be helpful if accurate). Only after this, the details like how this paper's method differs from Schlag et al., 2021, or how the presented idea is evaluated become easier to place into context.

Correlation between compositionality and generalization:
- The paper claims that the compositional structure seen in the intrinsic hypernet of MHA can explain the compositional generalization of transformers. However, there is a missing connection here. An important experiment is to evaluate with confidence estimates how the compositionality of attention correlates with the OOD generalization of the model.

Weak performance on language modeling:
- Despite the positive results on synthetic tasks, the paper's proposed method, HYLA, fails to outperform softmax attention for language modeling with a clear performance gap (Figure 6). The authors hypothesize that softmax is essential here, but find that HYLA with softmax results in a drop in performance (Table A1). Please look at the last question in the Questions section for a possible actionable avenue.

RMSNorm:
- It looks like RMSNorm at lines 179-180 is the operation from [1], but the paper is not cited.
- The authors state that it ensures the parameters of the value network are appropriately initialized to prevent exploding/vanishing gradients. There needs to be an explanation of why this should be the case.

References:
[1] Zhang, Biao and Sennrich, Rico. Root Mean Square Layer Normalization. NeurIPS 2019.

**Questions:**

HYLA with a single head:
- In the caption for Figure 5C, the authors state that the hypernet mechanism is absent for a single head. Why is this the case?

Scaling compositional difficulty:
- In Section 4.2, the authors state that 25% of possible rule combinations are held out for evaluation. How does performance change when this fraction is changed? Specifically, as the compositional difficulty (% of held-out rules) varies from 0% to 100% on the X-axis, how do the performances of softmax attention, linear attention, and HYLA on the Y-axis vary? This important experiment will help us understand how these methods compare in a fully in-distribution setting to much harder OOD settings.

Drop in performance of HYLA with softmax:
- For HYLA, in Eq (6), both the value network weights $\\mathbf{W}'\_{q,k}$ and $\\mathbf{W}\_{q,k}$ depend on $a\_{h,q,k}$, which makes sense from the hypernet perspective where $a$ is the latent code. In standard MHA, as shown in Eq (2), $\\mathbf{W}^\\text{out}$ operates after attention scaling has already been applied. However, from this perspective, Eq (6) indicates a re-application of attention scaling. Could this reapplication of attention scaling be responsible for the drop in performance when using softmax in Table A1, perhaps due to the reapplication of softmax being more prone to vanishing gradients?
- Do the language modeling results improve if a potential issue with using softmax with HYLA is identified and fixed?

---

> ### Author Response · Authors · 2024-11-21
>
> Thank you for having taken the time for an elaborate and insightful review. We appreciate the many suggestions for improvements you have made and believe that they have helped to improve the paper. We found your remark that you would like to rate the paper higher encouraging and hope that we were able to address your remaining concerns.
>
>
> > The proposed perspective of attention as hypernet is presented with the assumption of self-attention [...]
>
> Thank you for raising this concern, we are indeed focusing on self-attention in our experiments. The main derivation we present equally applies to cross-attention and we are clarifying this point in our updated version of the manuscript: In the case of cross-attention the keys and values come from the output of the encoder whereas the queries come from the output of the decoder. This means that based on the latent codes obtained from combining queries from the decoder and keys from the encoder, the hypernetwork mechanism generates a value network that processes inputs from the encoder stream and adds them to the decoder stream.
>
> > Clarity and connection of multi-head to compositionality [...]
>
> We agree that the mentioned paragraph was overly technical in the original submission and the important role of multiple heads was not sufficiently well communicated. We have rewritten it based on your feedback in our updated version of the manuscript hoping to have improved its clarity.
>
> > [...] An important experiment is to evaluate with confidence estimates how the compositionality of attention correlates with the OOD generalization of the model.
>
> It is difficult to operationalize the compositionality of attention directly and we would be curious if you had something else in mind: We can intervene on the compositionality of attention by removing all heads but one. With a single head (also see our response to your question in this regard), attention no longer supports the type of compositionality we hypothesize to help compositional generalization while maintaining the same number of parameters (we set $D^{\text{head}} = \frac{D}{H}$).
> The result of this intervention is shown in Figure 5C: In line with our hypothesis, for a single head the compositional generalization performance noticeably drops for all attention variants.
>
> > Despite the positive results on synthetic tasks, the paper's proposed method, HYLA, fails to outperform softmax attention for language modeling [...]
>
> > [...] Could this reapplication of attention scaling be responsible for the drop in performance when using softmax in Table A1, perhaps due to the reapplication of softmax being more prone to vanishing gradients?
>
> Following your suggestion, we have checked whether the ablation that combines the nonlinear value network of HYLA with the softmax normalization over keys of standard attention leads to vanishing gradients but have not observed this pathology. This is in line with other works that introduce additional multiplicative interactions inside of neural networks [e.g. 1,2] have similarly not reported instabilities in learning.
>
> One point we would like to stress is that we are introducing HYLA as an experimental device in an attempt to examine the hypothesis that the intrinsic hypernetwork of multi-head attention is what supports compositional generalization on abstract reasoning tasks that require recombining previously acquired operations. We take the fact that emphasizing this ability through HYLA improves language modeling performance compared to linear attention as evidence that the hypernetwork mechanism is also relevant in the context of language modeling. However we also acknowledge that softmax attention can be very flexible supporting other modes of operation such as for instance induction heads. HYLA was designed with compositionality in mind, but by emphasizing this aspect the layer might struggle with other behaviors that are expected from the attention layer in large complex tasks like language, as for example retrieval (which is not compositional).
>
> [1] Jayakumar et al. Multiplicative Interactions and Where to Find Them. (2019)
>
> [2] Chrysos et al. Deep Polynomial Neural Networks (2020)

---

> > ### Author Response · Authors · 2024-11-21
> >
> > > It looks like RMSNorm at lines 179-180 is the operation from [1], but the paper is not cited. [...]
> >
> > Thank you for pointing out the missing reference we have now added. We have also added a clarification that we are not using the learnable scalar RMSNorm introduces, instead simply normalizing by the standard root mean square of the attention weights across the head index. We have revised the main text with an explanation for how this helps to maintain the scale of a standard neural network initialization.
> >
> > > In the caption for Figure 5C, the authors state that the hypernet mechanism is absent for a single head. Why is this the case?
> >
> > In the case of a single head there is only a single possible value network that can be rescaled - but not composed with other value networks, i.e. in equation (4) the sum over h disappears. That is, while with multiple heads the hypernetwork consists of multiple possible value networks that can be composed with each other as specified through the latent code, in the single head case the latent code degenerates to a single scalar. We try to explain this point better in the second paragraph of section 4.4 corresponding to Figure 5C.
> >
> > > In Section 4.2, the authors state that 25% of possible rule combinations are held out for evaluation. How does performance change when this fraction is changed? [...]
> >
> > Indeed while we performed this experiment for the fuzzy logic task in Figure 2C, where we could observe performance of all attention variants eventually degrading in harder OOD settings, this was missing for SRAVEN. We have now performed the experiment you suggested for the SRAVEN task and added it to the appendix in Figure A5. Consistent with our other results, reinforcing the hypernetwork mechanism through HYLA seems to be beneficial across these settings before for high fractions of held-out rule combinations the performance of all models eventually degrades.

---

> > > ### Comment · Reviewer_59h3 · 2024-11-22
> > >
> > > Thank you for your responses! Most of my concerns and questions have been addressed.
> > >
> > >
> > > > It is difficult to operationalize the compositionality of attention directly and we would be curious if you had something else in mind
> > >
> > > Here's one potential idea. In the language emergence literature, topographical similarity is a popular metric to measure compositionality [1-5]. In [5], Figure 4(b) shows an example of showing how their measure of compositionality correlates with generalization. In your work, you already show that the latent codes exhibit meaningful distances in Figures 2 and 5, and SRAVEN's symbolic encodings can also be used to compute distances. With these, you can compute and report topographical similarities. These metrics can then be correlated with OOD performances that you also report throughout the paper. With these quantities, do we see that a higher topographic similarity results in higher OOD generalization, and vice versa? How confidently can we make any such claim?
> > >
> > > [1] Brighton and Kirby. Understanding linguistic evolution by visualizing the emergence of topographic mappings. Artificial life, 2006.
> > > [2] Lazaridou, Moritz Hermann, Tuyls, Clark. Emergence of linguistic communication from referential games with symbolic and pixel input. ICLR 2018.
> > > [3] Li and Bowling. Ease-of-teaching and language structure from emergent communication. NeurIPS 2019.
> > > [4] Chaabouni, Kharitonov, Bouchacourt, Dupoux, Baroni. Compositionality and Generalization In Emergent Languages. ACL 2020.
> > > [5] Ren, Guo, Labeau, Cohen, Kirby. Compositional languages emerge in a neural iterated learning model. ICLR 2020.
> > >
> > >
> > > > The main derivation we present equally applies to cross-attention and we are clarifying this point in our updated version of the manuscript
> > >
> > > Note: Footnote 1 in the current manuscript has a minor typo: "am" instead of "an".

---

> > > > ### Author Response · Authors · 2024-11-25
> > > >
> > > > We are happy to hear that we were able to address most of your questions and concerns. As for your remaining point:
> > > >
> > > > > Here's one potential idea. In the language emergence literature, topographical similarity is a popular metric to measure compositionality [1-5]. In [5], Figure 4(b) shows an example of showing how their measure of compositionality correlates with generalization. In your work, you already show that the latent codes exhibit meaningful distances in Figures 2 and 5, and SRAVEN's symbolic encodings can also be used to compute distances. With these, you can compute and report topographical similarities. These metrics can then be correlated with OOD performances that you also report throughout the paper. With these quantities, do we see that a higher topographic similarity results in higher OOD generalization, and vice versa? How confidently can we make any such claim?
> > > >
> > > > Thank you for bringing the work on topographical similarity in the language emergence literature to our attention. We appreciate the suggestion for an additional analysis based on it.
> > > >
> > > > As you outline, measuring topographical similarity in our setup would entail correlating distances between SRAVEN tasks with distances between latent codes. While this is an interesting idea, there are several challenges to instantiating this analysis. Most importantly, since we are not operating in a language setting, we would need to define a distance metric on the space of SRAVEN tasks. Unfortunately, we do not believe a clear choice exists here. A SRAVEN task is defined by the combination of rules and a permutation across features. Let’s for the moment neglect the permutation and simply try to define a distance on the combination of rules. One possibility could be to adopt the Levenshtein/edit distance between two SRAVEN tasks often used in language. With this choice and a simple encoding that assigns each rule a unique symbol (e.g. A: progression +1 B: progression +2, C: distribute-three, …), we would assert that any two rules are equidistant and imply that a +1 progression rule would have the same distance to a +2 progression rule as it would have to a distribute-three rule. Intuitively these distances should be different so we might be enticed to come up with a different symbolic encoding or maybe a different distance altogether. This is not to say it is impossible but that there are potentially many possible choices which most likely strongly affect the final outcome.
> > > >
> > > > That being said, your original question was  "how the compositionality of attention correlates with the OOD generalization of the model.". Our paper contains several interventions which we believe provide evidence for this point. Specifically, we intervene on the hypernetwork mechanism we posit to underlie the compositionality of attention and find that:
> > > >
> > > > - Reinforcing the hypernetwork mechanism through HYLA (e.g. Figure 2C, Figure 4) or to varying degrees in our model ablations (Table A1) improve OOD performance
> > > > - Disrupting the hypernetwork mechanism by using a single head vs. multiple heads impairs OOD performance (Figure 5C)
> > > >
> > > > In addition, we find the hypernetwork latent code to be functionally structured and predictive of the task components (Figure 2BD and Figure 5BEF). Together, we believe these experiments provide evidence for the hypothesis that the hypernetwork mechanism present in multi-head attention supports compositional generalization.
> > > >
> > > > We hope that these considerations have helped to resolve your remaining point adequately. Thank you again for deeply engaging with our work.

---

> ### Comment · Reviewer_59h3 · 2024-11-26
> **Thank you**
>
> Thank you for all your responses! My primary concerns have been addressed, and I have increased my score.

---

### Official Review · Reviewer_ZLCD · 2024-11-07

**Soundness:** 3
**Presentation:** 3
**Contribution:** 3
**Rating:** 8
**Confidence:** 3

**Summary:**

This work reformulates linear multi-head attention as a hypernetwork to study compositional generalisation and introduces a simple non-linear variant that substantially increases generalisation both on abstract symbolic tasks as well as language modeling.

**Strengths:**

The paper is well-written and studies an important problem, namely compositional generalisation, from a new angle and with promising empirical results. The writing, structure and related work is outlined clearly, and (almost) all relevant details with regards to the theory and the empirical experiments are available in either the main text or the appendix. The perspective to see multi-head attention as a hypernetwork is novel and might open interesting future avenues for study.

**Weaknesses:**

One problem in the empirical evaluation is a certain lack of control over whether the suggested modification (HYLA) is actually increasing compositional generalization or whether it's just a way of increasing the capacity of the model. To this end, it would be interesting to look at the training loss of the models: does HYLA reach the same minimum as linear / softmax attention, or is it already performing much better on the training set (suggesting it is more the capacity increase of the model rather than a genuine way of increasing compositional generalisation abilities).

Similarly, it didn't get clear to me how the OOD sets were sampled? E.g., in the fuzzy logic dataset, when you say "hold out a fraction of the combinations for OOD" (line 206), does this refer to combinations of K terms, or are all K terms sampled and you just subsample each (meaning, in the OOD set all combinations of K are present)?

**Questions:**

- In HYLA, you not only add the nonlinearity but also change the normalisation. Did you perform ablations w.r.t. the normalization, meaning how well does HYLA perform with the standard normalization (or vice versa, if you add RMSHead in linear attention)?

- Please report the training performance of the models to shed light on the generalisation vs capacity question.

- Please provide more details on the OOD splitting of the datasets.

---

> ### Author Response · Authors · 2024-11-21
>
> Thank you for your positive review highlighting that you found the paper to provide a novel perspective on an important problem. Your review helped us clarify an important point on the possibility of HYLA mereling increasing the capacity of the model rather than specifically improving compositional generalization in the updated version of the manuscript (and in our response below of course).
>
> > One problem in the empirical evaluation is a certain lack of control over whether the suggested modification (HYLA) is actually increasing compositional generalization or whether it's just a way of increasing the capacity of the model. To this end, it would be interesting to look at the training loss of the models: does HYLA reach the same minimum as linear / softmax attention, or is it already performing much better on the training set (suggesting it is more the capacity increase of the model rather than a genuine way of increasing compositional generalisation abilities).
>
> Thank you for raising this important point. We have added the training loss on the fuzzy logic  tasks corresponding to Figure 2C in appendix figure A3D to complement the training R2 score we already reported in the appendix of the original submission. Similarly the new appendix figure A4 shows the training loss on the SRAVEN scaling experiments of Figure 4. We will point more prominently to these in the main text.
>
> We generally find no consistent differences between methods in terms of in-distribution training loss. Together with the property that HYLA has exactly the same number of parameters as linear and softmax attention, we take this as an indication that it specifically improves compositional generalization and is not simply a way of improving overall model capacity.
>
> > Similarly, it didn't get clear to me how the OOD sets were sampled? E.g., in the fuzzy logic dataset, when you say "hold out a fraction of the combinations for OOD" (line 206), does this refer to combinations of K terms, or are all K terms sampled and you just subsample each (meaning, in the OOD set all combinations of K are present)?
>
> Apologies for the confusion, we will try to clarify this point in the manuscript: In the fuzzy logic benchmark, each task is defined by a combination of K terms. So for instance if we have 3 possible terms A, B, C, and K=2 then the possible tasks are AB, AC, BC. During training we only train on a subset of tasks, e.g. we train on AB, BC and during OOD evaluation we evaluate on the held-out task, e.g. AC.
>
> > In HYLA, you not only add the nonlinearity but also change the normalisation. Did you perform ablations w.r.t. the normalization, meaning how well does HYLA perform with the standard normalization (or vice versa, if you add RMSHead in linear attention)?
>
> Yes, we perform a number of ablations in Appendix A1 including removing the RMSHead normalization from HYLA and adding it to linear attention. We find that while simply applying RMSHead to linear attention (Linear Attention + RMSHead) and also the attention-weighted sum operation without nonlinearity (HYLA − nonlinearity) both by themselves can slightly boost performance, the full HYLA model generally performs best in terms of compositional generalization. We will point more prominently to the ablations in the main text.

---

### Author Response · Authors · 2024-11-21

We thank all reviewers for their positive reviews and insightful comments. We have revised the manuscript based on your feedback to improve the clarity of the exposition (see individual responses) and have added new results and figures based on several experiments suggested by the reviewers.

We believe these additional results provide further evidence for the central hypothesis of the paper that the hypernetwork mechanism present in multi-head attention supports compositional generalization.

Notably, we added the following to the revised version of the manuscript:

* As suggested by Reviewer ZLCD, we added appendix figures A3 and A4 on page 18-19 that show the training loss on all fuzzy logic and SRAVEN experiments conducted in the main text. They provide further evidence that encouraging the hypernetwork mechanism identified in multi-head attention does not simply result in additional capacity to fit the training data but specifically leads to improvements in compositional generalization.
* As suggested by Reviewer 59h3, we conduct a new experiment that compares the compositional generalization performance as the fraction of OOD tasks varies for SRAVEN in Figure A5 on page 19 similar to the existing experiment on the fuzzy logic task in Figure 2C. The experiment confirms that encouraging the hypernetwork mechanism improves compositional generalization even for large fractions of held-out tasks before performance eventually degrades across models.
* As suggested by Reviewer yrk8, we have conducted new experiments to evaluate to what extent replacing the MLP layer of the transformer block with Mixture of Experts (MoE) improves compositional generalization in Appendix A1 and Table A1. We find only moderate improvements in line with our findings on how increasing model capacity in general benefits compositional generalization in Figure 4.
* As suggested by Reviewer 47bR, we perform additional model ablations varying the nonlinearity of the value network introduced in HYLA (see individual response below) showing that encouraging the hypernetwork mechanism with other nonlinearities leads to similar improvements in compositional generalization.

We hope to have resolved your concerns (please see the individual responses) and are looking forward to engaging in further discussions.

---

### Meta-Review · Area_Chair_SXjo · 2024-12-20

**Metareview:**

This paper demonstrates a mathematical equivalence between multihead attention and linear hypernetworks, which is then used to provide an explanation for why Transformers are able to compositionally generalize to some extent. The attention scores per head can be interpreted as a low-dimensional latent code that predicts the resulting linear value network that processes the input. Then, this insight is used to construct a more expressive approach to compositional processing by modifying MHA to make the value network nonlinear, and also introduce a normalization scheme that leads to linear value networks that are more consistent with desirable network initialization properties. Through experiments on two toy reasoning datasets (a fuzzy boolean logic task and a symbolic Raven matrices task), the paper shows evidence of compositional generalization that is enhanced for the nonlinear version, lending support for the hypernetwork view of attention. Qualitative visualizations show that the heads are consistent with hypothesized compositional functions. Finally, experiments on language modeling show that the nonlinear hypernetwork approach improver over standard linear attention baselines, adding further support for the hypernetwork view of attention.

Strengths:
Well-written paper posing an interesting hypothesis with provocative results on compositional generalization. May open new design space choices for improving attention schemes in general and towards better generalization in reasoning tasks. The nonlinear modification and normalization scheme of HYLA may be valuable in its own right as a drop-in replacement for attention. The paper may also spur new ways of approaching interpretability in attention models. Experiments are nice in that they allow controlling degree of compositionality.

Weaknesses:
Not really a core weakness, but it would be nice to see controlled experiments with cross-attention, since it was claimed that the same treatment should hold. There are also some unexplained aspects of standard softmax self-attention that are not captured by the current treatment of linear MHA as a hypernetwork (and the HYLA variant), and one reviewer is not fully convinced that the hypernetwork equivalence is a useful construct to explain compositional generalization in softmax attention.

Decision reasoning:
Reviewers unanimously felt the paper is a strong contribution, with 4/5 reviewers giving scores of 8. I agree with the reviewers’ assessments and find the paper quite interesting, particularly towards better understand why Transformers are able to reason compositionally in at least some cases. The authors also put a lot of effort into addressing reviewer concerns, which made the final paper significantly stronger. Papers like this are useful for understanding methodology, but may also lead to new developments — while the authors caution that the HYLA variant of MHA is more of an ‘experimental device’ rather than a legitimate enhancement, it may inspire new advances that enhance the expressiveness of attention.

**Additional Comments On Reviewer Discussion:**

Reviewers were generally encouraging about the paper from the start but brought up several important weaknesses that were addressed by the authors through clarification and additional experiments. The changes were well received and were also easy for me to follow to be able to make my assessment.

---

### Decision · Program_Chairs · 2025-01-22

Accept (Oral)